

# The effect of agmatine on trichothecene type B and zearalenone production in *Fusarium graminearum, F. culmorum* and *F. poae*

Matias Pasquali, Emmanuelle Cocco, Cédric Guignard and Lucien Hoffmann

Department of Environmental Research and Innovation, Luxembourg Institute of Science and Technology (LIST), Belvaux, Luxembourg

## ABSTRACT

Agmatine and other putrescines are known for being strong inducers of deoxynivalenol (DON) production in *Fusarium graminearum*. Other important species produce DON and/or other trichothecene type B toxins (3 acetylated DON, 15 acetylated DON, Fusarenon-X, Nivalenol), such as *F. culmorum* and *F. poae*. In order to verify whether the mechanism of the regulation of trichothecene type B induction by agmatine is shared by different species of Fusarium, we tested the hypothesis on 19 strains belonging to 3 Fusarium species (*F. graminearum, F. culmorum, F. poae*) with diverse genetic chemotypes (3ADON, 15ADON, NIV) by measuring trichothecene B toxins such as DON, NIV, Fusarenon-X, 3ADON and 15ADON. Moreover, we tested whether other toxins like zearalenone were also boosted by agmatine. The trichothecene type B boosting effect was observed in the majority of strains (13 out of 19) in all the three species. Representative strains from all three genetic chemotypes were able to boost toxin production after agmatine treatment. We identified the non-responding strains to the agmatine stimulus, which may contribute to deciphering the regulatory mechanisms that link toxin production to agmatine (and, more generally, polyamines).

## INTRODUCTION

Mycotoxin regulation mechanisms leading to accumulation of toxins in the plant, and consequently in grains used for human and animal consumption, are still partially unknown. Nutrients and specific molecules are supposed to play a key inducing role in activating toxin pathways *in planta*. *Fusarium graminearum* Schwabe [teleomorph *Gibberella zeae* (Schwein.) Petch], *Fusarium culmorum* (WG Smith) Sacc. and *Fusarium poae* (Peck) Wollenw. are the main species associated with trichothecene type B (TB) production in Fusarium Head Blight (FHB) in wheat in different agricultural areas in the world. The most important Fusarium TB are deoxynivalenol (DON), the amount of which in food and feed is controlled in many countries, 3-acetylated DON (3ADON), 15-acetylated DON (15ADON), Nivalenol (NIV) and Fusarenon X (FUSX). *F. graminearum*

Corresponding author
Matias Pasquali,
matias.pasquali@list.lu,
matias.pasquali@gmail.com

and *F. culmorum* can also produce Zearalenone (ZEA), a toxin which is legislatively regulated in food and feed due to its estrogenic effect on humans and animals. Each Fusarium strain is able to produce some of these toxins depending on the set of genes present in the genome (*Proctor et al., 2009*) and the triggering factors leading to toxin production. Because toxigenic risk in food depends on multiple factors including the type of fungal population colonizing the plant (and its toxigenic potential), much effort has been devoted to studying the epidemiology of different chemotypes worldwide (*Pasquali & Migheli, 2014*), using genetic means to discriminate populations that are more or less toxigenic (*Von der Ohe et al., 2011*). Chemical determination of chemotypes is still an important approach to confirm genetic analysis based on gene polymorphisms (*Desjardins, 2008*); therefore, liquid media able to stimulate toxin production have been widely employed for determining the toxin potential of isolates.

Mechanisms of toxin induction in *F. graminearum* include sugar types (*Jiao, Kawakami & Nakajima, 2008*), pH effects (*Gardiner, Kazan & Manners, 2009*; *Merhej et al., 2010*), inorganic compounds (*Tsuyuki et al., 2011*; *Pinson-Gadais et al., 2008*), oxidative stress (*Ponts et al., 2007*), fungicides (*Magan et al., 2002*), light (*Kim, Son & Lee, 2014*), and water activity levels (*Llorens et al., 2004*; *Schmidt-Heydt et al., 2011*) and have also been linked to chemotype diversity (*Ponts et al., 2009*). *F. graminearum* boosts toxin production when grown in a medium with agmatine and other putrescines to levels that are comparable to the high contaminations observed *in planta* (*Gardiner et al., 2009*). Hypotheses on the role of polyamines *in planta* as a cue for the production of trichothecene mycotoxins by *F. graminearum* during the FHB disease have been formulated (*Gardiner et al., 2010*) and the use of inhibitors of polyamine transport in the fungal cells have been proposed as a novel approach to limit toxin contamination in grains (*Crespo-Sempere et al., 2015*). Studying the effect of agmatine on other species can elucidate whether this mechanism is also effective in other *Fusarium* species that are often found to coexist in agricultural settings (*Giraud et al., 2010*), directly contributing to toxin accumulation (*Beyer et al., 2014*). Because the colonization of the plant by the fungus and the resulting production of toxins are the outcomes of the interaction of the environment, the fungus and the plant, here we focused on how fungal diversity is affected by a putative plant-derived inducer of toxin production. We therefore tested the level of toxin induction in relation to fungal diversity in 3 *Fusarium* species when confronted with 2 standardized media which are mild and strong inducers of toxin synthesis.

Therefore, the aims of the work were to evaluate the toxigenic potential *in vitro* of a set of isolates and to analyse the effect of agmatine as toxin-inducing compounds across species and chemotypes.

## MATERIAL AND METHODS

### Isolates and growth conditions

Isolates used in this study are listed in Table 1. As determined in our laboratory, they have a different geographic origin and belong to different genetic chemotypes (*Pasquali et al., 2011*). Strains are conserved in the Luxembourg Microbial Culture Collection

**Table 1** Strain identification code, species, genetic chemotype, year of isolation, geographical origin, strain collection where the strain is deposited.

| Strain identification code | Species | Genetic chemotype | Year of isolation | Geographical origin | Strain collection[a] |
|---|---|---|---|---|---|
| 13-01 | *F. culmorum* | 3ADON | 2008 | (Hoscheid) Luxembourg | LuxMCC |
| 233 | *F. culmorum* | NIV | 2007 | (Reisdorf) Luxembourg | LuxMCC |
| 189 | *F. culmorum* | 3ADON | 2007 | (Reisdorf) Luxembourg | LuxMCC |
| 01-02 | *F. culmorum* | NIV | 2008 | (Kehlen) Luxembourg | LuxMCC |
| 557 | *F. culmorum* | 3ADON | 2007 | (Reuler) Luxembourg | LuxMCC |
| 708 | *F. culmorum* | NIV | 2007 | (Christnach) Luxembourg | LuxMCC |
| MUCL555 | *F. culmorum* | NIV | 1952 | Unknown | MUCL |
| MUCL15500 | *F. culmorum* | NIV | 1946 | Netherlands | MUCL |
| MUCL11946 | *F. graminearum* | 3ADON | 1969 | Belgium | MUCL |
| NRRL37099 | *F. graminearum* | 3ADON | 1994 | (Manitoba) Canada | NRRL |
| MUCL42825 | *F. graminearum* | NIV | 2000 | Belgium | MUCL |
| 16-09 | *F. graminearum* | NIV | 2008 | (Troisvierges) Luxembourg | LuxMCC |
| 734 | *F. graminearum* | 15ADON | 2007 | (Christnach) Luxembourg | LuxMCC |
| 11-24 | *F. graminearum* | 15ADON | 2008 | (Echternach) Luxembourg | LuxMCC |
| UMW00-706 | *F. graminearum* | 15ADON | >2000 | USA | Courtesy of L. Gale |
| 80 | *F. poae* | – | 2007 | (Nothum) Luxembourg | LuxMCC |
| 491 | *F. poae* | – | 2007 | (Nothum) Luxembourg | LuxMCC |
| 57B | *F. poae* | – | 2007 | (Kayl) Luxembourg | LuxMCC |
| 504A | *F. poae* | – | 2007 | (Kayl) Luxembourg | LuxMCC |

**Notes.**

[a]LuxMCC, Luxembourg Microbial Culture Collection (LUX); MUCL, *Mycothèque* de l'Universite Catholique de Louvain (BEL); NRRL, Agricultural Research Service culture collection (USA).

accessible at www.luxmcc.lu (*Piec et al., 2016*). All isolates were grown on V8 (V8 juice 20%, $CaCO_3$ 2 g, 18 g agar, $H_2O$ to 1 L) plates for 4 days at 20 °C in the dark. The growing edge of the mycelium was divided into 2 mm squares and one square each was used to inoculate 10 ml media in sterile glass tubes. The assay was carried out with three completely independent biological replicates.

In order to analyse toxin induction, 2 media were used and compared. The first medium (J), from *Jiao, Kawakami & Nakajima (2008)* (containing 1 g $K_2HPO_4$, 0.5 g KCl, 0.5 g $MgSO_4 \cdot 7H_2O$, 2 g L-glutamic acid, 10 mg $FeSO_4 \cdot 7H_2O$, 10 g sucrose in a litre of solution) was compared to a medium (G) from *Gardiner et al. (2009)*, containing (30 g sucrose, 1.15 g Agmatine, 1 g $KH_2PO_4$, 0.5 g $MgSO_4 \cdot 7H_2O$, 0.5 g KCl, 10 mg $FeSO_4 \cdot 7H_2O$ in a litre solution). The two media have approximately the same C/N ratio, which is known to play a role in toxin regulation (*Hestbjerg et al., 2002*) but differ for the source of nitrogen and the amount of carbon source. Tubes were shaken at 180 rpm in the dark for 10 d. Fungal biomass was then filtered and weighted with a precision balance after drying using a freeze drier for 48 h, and the remaining media were kept for further toxin extraction.

## Chemical analysis

The medium was filtered through a 0.2 $\mu$m GHP membrane filter (PAL, MI, USA) and diluted in methanol (extract/methanol, 9/1, V/V) in order to be in the appropriate solvent ratio for chromatographic analysis. The quantification of all mycotoxins except 3-ADON and 15-ADON was performed by LC coupled to tandem mass spectrometry (LC-MS/MS, Dionex Ultimate 3000, AB/Sciex API 3200, Foster City, CA, USA) in multiple reaction monitoring (MRM) in positive/negative switching mode. The LC analytical column was an Agilent Zorbax Eclipse Plus $C_{18}$ (2.1 $\times$150 mm, 3.5 $\mu$m) with a mobile phase consisting of methanol and water containing 2.5 mM of ammonium acetate in a linear gradient. The specific analysis of 3-ADON and 15-ADON was achieved using a second LC-MS/MS method on an Agilent 1260 LC coupled to an AB/Sciex 4500 QTrap mass spectrometer. The column was an Agilent Poroshell 120 EC-$C_{18}$ (2.1 $\times$ 150 mm, 2.7 $\mu$m) and the eluents were the same as for the first method. All mycotoxins were quantified by external calibration based on pure standards (Biopure, Tulln, AT and Sigma-Aldrich, Schnelldorf, Germany). The detection and quantification limits were 1.5 ng/ml of liquid culture for all toxins. The analyses were done in two technical replicates and the average value was considered for each biological replicate.

## Statistical analysis

All data were analysed using PASW version 19 and SigmaPlot version 12.5. Overall TB production was calculated summing up all trichothecenes type B measured (DON, 3ADON, 15ADON, NIV, FUS-X). Mass-corrected toxin concentrations were calculated and used for determining the effect of nitrogen source on toxin boosting. The Mann–Whitney Rank Sum test was used to verify the effect of the medium on masses and the effect of nitrogen source in the medium on mass-corrected summed trichothecene type B values. The Kruskal–Wallis One-Way Analysis of Variance on Ranks was used to verify the effect of the medium on the average ratio of TB production in the two media for each strain classified either as species or as chemotypes. Significant differences are considered when $p < 0.05$.

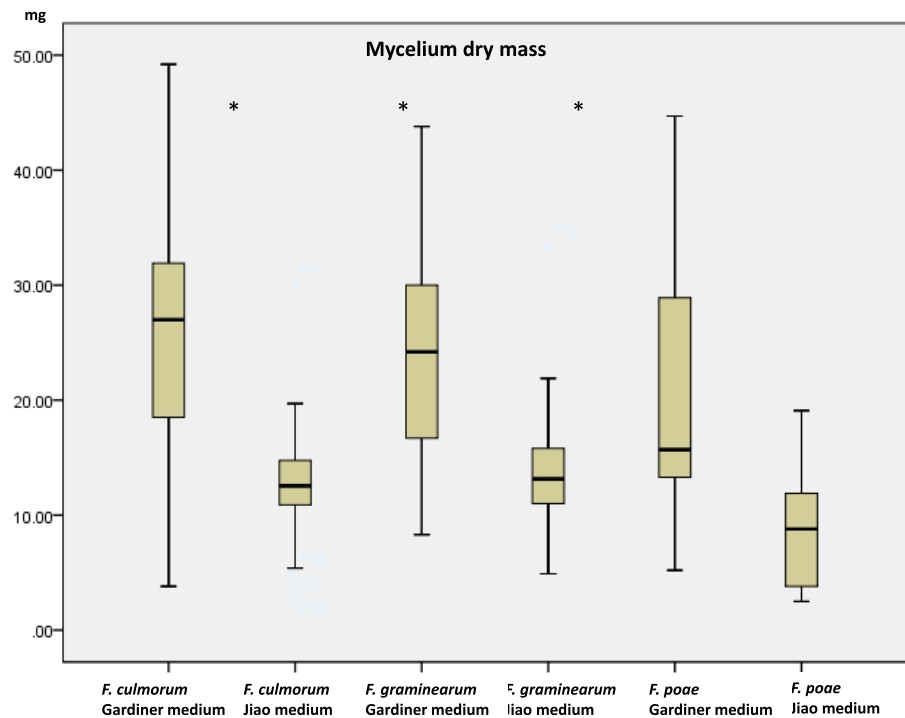

**Figure 1** **Dry mycelium mass measured in the two media in the three species.** Mass in mg of dried mycelium grown 10 days in the two media (Gardiner medium containing agmatine; Jiao medium containing glutamic acid). Asterisks indicate significant differences using Mann–Whitney rank test ($p < 0.05$).

## RESULTS AND DISCUSSION

Growth (dry mass) and toxin production using a multimethod assay including DON, 3ADON, 15ADON, NIV, FUS-X, ZEA, T2, HT2 were measured on a set of 19 strains (Table 1) when grown in the two media. Mycelium growth (estimated as dry mass) was influenced by the medium increasing, as expected, in the high containing saccharose medium ($p < 0.001$). The mass of the strains in glutamic acid medium ranged from 3.4 mg to 19.6 mg while in the agmatine/saccharose-rich medium, masses ranged from 8.9 to 36.1 mg. Growth results are comparable with results obtained by *Jiao, Kawakami & Nakajima (2008)* for the medium with glutamic acid. We observed that FG and FC had a similar range of growth in each medium while FP differed significantly in growth compared to the other two species ($p > 0.05$). No effect of the chemotype ($p = 0.328$) could be identified. In all three species there was nonetheless a significant growth effect caused by the medium being the agmatine/high saccharose medium, a booster of mass growth ($p > 0.001$ for all the three comparisons, Fig. 1). The overall increase in dry mass caused by the agmatine medium can be attributed to the higher amount of saccharose in the medium, with carbon being the main constituent of fungal biomass (*Newell & Statzell-Tallman, 1982*). To evaluate toxin production, we therefore normalized the toxin produced by the dry mass per ml.

The two sources of nitrogen had an overall significant effect on trichothecene type B mass-corrected production across the three species ($p = 0.003$). The agmatine medium induced higher TB production in 13 out of 19 strains. Six isolates produced average TB toxins in ng/ml corrected by their mass in mg above 150 ng/ml/mg (*F. culmorum* 233, 13-01; *F. graminearum* 734, 37099 and MUCL42825 and *F. poae* 80). All these isolates increased their production in the medium containing agmatine with a boosting factor ranging between 192X of 13-01 to 15X of isolate 80. Four isolates (01-02, 557. MUCL15500, UMW00706) demonstrated the opposite behaviour when grown in the medium containing agmatine (Fig. 2).

We could also confirm that strains with a determined genetic chemotype can also produce a minor amount of the other acetylated and non-acetylated trichothecenes (*De Kuppler et al., 2011*). Nonetheless, there was a good correspondence between the genetic chemotype and the major toxin type produced (Table S1).

By analysing the average ratio of toxin production in the isolates according to their species or their chemotype, we could not detect a clear effect of any of the two categories. In fact, the response of toxin production to the two media was not significantly affected by the species ($p = 0.552$) nor by the chemotype ($p = 0.578$). A previous hypothesis suggesting differences at the chemotype level on toxin regulation (*Ponts et al., 2009*) may have been biased by the limited number of strains used.

ZEA production, in the only consistent producer of the lot (*F. culmorum* 01-02), doubled in the agmatine medium suggesting that the mechanism boosting TB production also potentially affects ZEA production. This would imply that the agmatine regulation of ZEA and of the trichothecene cluster is common, despite gene expression data from the available microarray studies not showing consistent concordant patterns (*Sieber et al., 2014*). A larger set of isolates producing ZEA is nonetheless needed to verify this hypothesis.

In our analysis, we also included two food and feed monitored trichothecene type A (TA) toxins but no isolates of *F. poae* produced any detectable amount of T2 and HT-2. As the biosynthesis of TAs follows a similar pathway (*Kimura et al., 2007*), we can expect similar effects due to agmatine on TA. However, we could not confirm this with the isolates used in our study.

## CONCLUSIONS

The effect of 2 different media containing agmatine or glutamic acid on toxin induction and growth in a total of nineteen strains belonging to the *F. graminearum (n = 7)*, *F. culmorum (n = 8)* and *F. poae (n = 4)* species was evaluated. With our species-comparative study, we confirmed that on average the mechanism of triggering toxin production by agmatine is confirmed in the 3 species but that the mechanism is also significantly influenced by strain diversity (*Pasquali et al., 2015*). The identification of isolates that do not respond to agmatine may help decipher the pathways leading to specific agmatine regulation in the fungus. Despite these data being obtained by culturing the fungus *in vitro*, they can contribute to explaining the partial diversity of toxin contamination observed *in planta*: indeed, on average, different cultivars accumulate different

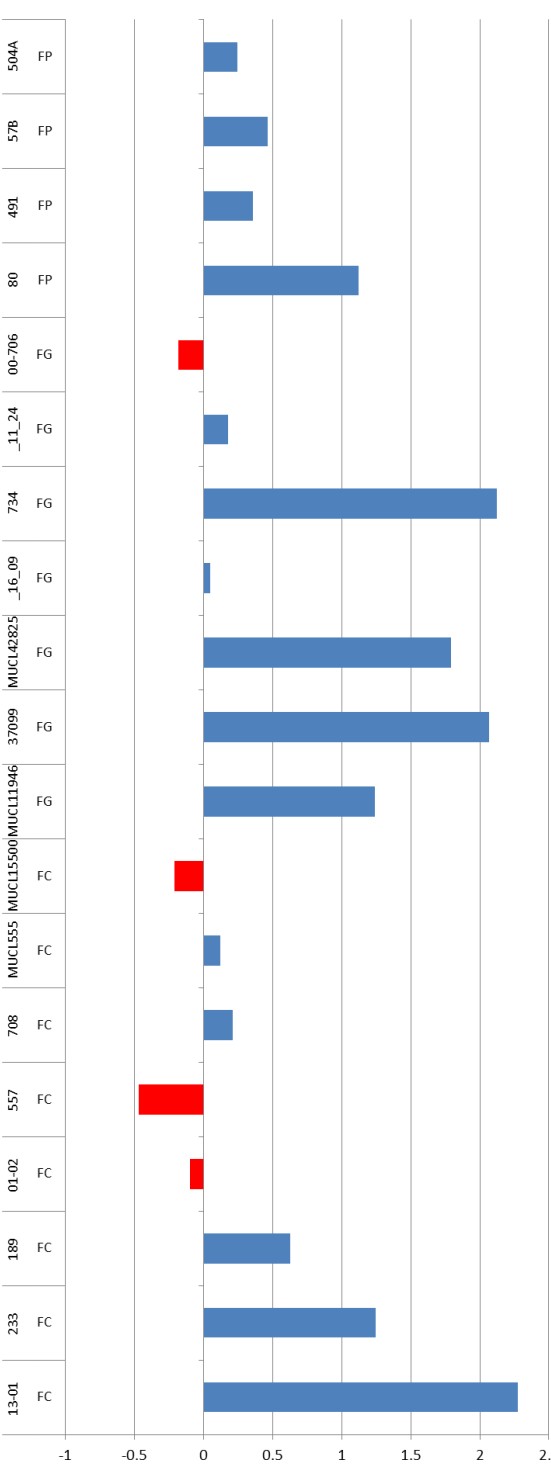

**Figure 2 Toxins ratio comparing the two media.** Log 10 transformed average ratio of TB production in agmatine vs glutamic acid medium. Values are the sum of all trichothecene type B detected (DON, 3ADON, 15ADON, NIV, FUSX). In red, isolates that have higher amount of TB in the glutamic acid medium while in blue those that have higher amount in the agmatine medium.

levels of DON (*Ji et al., 2015*). Hypotheses on the effect of the plant oxidative status (*Waśkiewicz et al., 2014*) and of polyamine concentration on toxin synthesis *in planta* have been formulated (*Gardiner et al., 2010*). Here we showed that strain diversity can also account for a factor 1000X  in the level of toxin accumulation. Exploring the diversity of interactions between strains and toxin triggering compounds such as polyamines (*Valdes-Santiago et al., 2012*) is fundamental to identifying general inhibitors of TB accumulation in the plant.

## ACKNOWLEDGEMENTS

We thank Laurence Joly for her technical assistance with toxin measures and Marine Pallez for her support in fungal collection maintenance. Lindsey Auguin is acknowledged for the English revision of the manuscript.

### Funding

We received financial support from the Administration des Services Techniques de l'Agriculture du Luxembourg. The funders had no role in study design, data collection and analysis, decision to publish, or preparation of the manuscript.

### Grant Disclosures

The following grant information was disclosed by the authors:
Administration des Services Techniques de l'Agriculture du Luxembourg.

### Competing Interests

The authors declare there are no competing interests.

### Author Contributions

- Matias Pasquali conceived and designed the experiments, performed the experiments, analysed the data, wrote the paper, prepared figures and/or tables, reviewed drafts of the paper.
- Emmanuelle Cocco performed the experiments, analysed the data, wrote the paper, prepared figures and/or tables, reviewed drafts of the paper.
- Cédric Guignard contributed reagents/materials/analysis tools, wrote the paper, prepared figures and/or tables, reviewed drafts of the paper.
- Lucien Hoffmann contributed reagents/materials/analysis tools, wrote the paper, reviewed drafts of the paper.

### Data Availability

The raw data has been supplied in Data S1.

### Supplemental Information

Supplemental information for this article can be found online at http://dx.doi.org/10.7717/peerj.1672#supplemental-information.

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
