# Peer review of "The effect of agmatine on trichothecene type B and zearalenone production in Fusarium graminearum, F. culmorum and F. poae"

_PeerJ, doi:10.7717/peerj.1672_

## Round 0.1 · original submission · Major Revisions

Dear Author,

Thank you very much for submission of your paper to Peer J. Though you did good work as presented in the paper I agree with the comments made by reviewer 2. So please re-submit your paper after suitable corrections as mentioned by both reviewers.

Best regards,
Vijai Kumar Gupta

Reviewer 1 ·

Basic reporting

no comments, included in comments to authors

Experimental design

no comments, included in comments to authors

Validity of the findings

no comments, included in comments to authors

Additional comments

In the present paper, the authors want to shed light into the regulatory network of trichothecenes by investigating the importance of specific AA in this process. I have several major comments that I feel are overlooked by the authors, and therefore, the paper is not acceptable for publication.
1. In the abstract, the authors say that other species produce DON and other type B trichothecenes, to my knowledge, F. poae is not a DON producer.
2. It is well known that F. poae produces type A and type B trichothecenes (see paper by Moretti and coworkers). This means that when you only look at a specific subset of toxins, you can never have an idea of the total titer of trichothecenes produced. In this way, you should also verify DAS, NEO,...
3. In the introduction, the authors mention inducers of trichothecenes, still this list is very short, other factors that are known to influence this are not mentioned e.g. light, fungicides, aw value, ...all aspects that have been addressed by other research groups.
4. What is meant with a "moderate toxin producer", this totally not scientific. What do the authors mean with moderate.
5. I do not understand why all the toxins are summed up. especially ZEN which is a structurally totally different toxin. Also the baseline production of trichothecenes (ppm levels) is completely different from that of ZEN (ppb range). So the picture represented is not correct.
6. for me, it is not clear whether the toxin data were calculated taking into account the fungal biomass.
7. statistics: how can you investigate variance and normality by box and Q-Q plot, the authors should use statistical tests. Figures illustrate already that variances are not equal between isolates, so all data should have been investigated using a non-parametric approach

Reviewer 2 ·

Basic reporting

Manuscript entitle "The effect of agmatine on trichothecene type B and zearalenone production in Fusarium graminearum, F. culmorum and F. poae" well written and experimental data presented very nicely.

Experimental design

Well design experiment with data analysis.

Validity of the findings

no comments

Annotated reviews are not available for download in order to protect the identity of reviewers who chose to remain anonymous.

---

## Round 0.2 · Minor Revisions

Dear Author,

Your article is accepted pending some remaining edits to the English language.

Best regards,
Vijai Kumar Gupta

Reviewer 2 ·

Basic reporting

No comments

Experimental design

No Comments

Validity of the findings

No comments

Additional comments

The Authors made all the suggested corrections in the manuscript. So, I would like to "ACCEPT" this article.

Reviewer 3 ·

Basic reporting

The effect of agmatine on trichothecene type B and zearalenone production in Fusarium graminearum, F. culmorum and F. poae is suitable for publication.

I found few grammatical / typological errors.

Experimental design

The statistical data analysis is complete

Validity of the findings

The paper could be strengthened with additional details in the results and discussion, but it is acceptable as it is.

Additional comments

Manuscript accepted for publication.

---

## Round 0.3 · accepted · Accept

Thank you for the English language edits to the revised manuscript, which is now up to the standard of PeerJ and will be informative to readers. I recommend the manuscript for Publication.